# Q*Agent: Optimizing Language Agents with Q-Guided Exploration

## Abstract

Language agents have become a promising solution to complex interactive tasks. One of the key ingredients to the success of language agents is the reward model on the trajectory of the agentic workflow, which provides valuable guidance during training or inference. However, due to the lack of annotations of intermediate interactions, most existing works use an outcome reward model to optimize policies across entire trajectories. This may lead to sub-optimal policies and hinder the overall performance. To address this, we propose Q*Agent, leveraging an estimated Q value to generate intermediate annotations for open language agents. By introducing a reasoning tree and performing process reward modeling, Q*Agent provides effective intermediate guidance for each step. This guidance aims to automatically annotate data in a step-wise manner. Besides, we propose a Q-guided ~~exploration~~generation strategy that can significantly boost model performance by providing process guidance during inference. Notably, even with almost half the annotated data, Q*Agent retains strong performance, demonstrating its efficiency in handling limited supervision. We also empirically demonstrate that Q*Agent can lead to more accurate decision making through qualitative analysis.

## 1 Introduction

Open-source language models rely on supervised fine-tuning (SFT) to accomplish complex agent tasks (Chen et al., 2023; Yin et al., 2024). However, the substantial human annotations required for collecting training data present a significant bottleneck, limiting both performance and scalability. This challenge is particularly pronounced in agent tasks (Yao et al., 2022; Shridhar et al., 2021; Wang et al., 2022), where data scarcity is a critical issue due to the inherent complexity and diversity of the tasks. Collecting high-quality training data for such tasks often involves intricate, context-specific interactions, which demand expert knowledge and extensive effort. To overcome this challenge, self-improvement techniques have emerged as a promising area of research (Wang et al., 2024a; Singh et al., 2023; Hosseini et al., 2024; Zhang et al., 2024), enabling models to learn from self-generated data without extensive human intervention. A central question in this paradigm is how to better and more efficiently explore useful trajectories that can enhance the model's capabilities.

An essential component in self-improvement methods is the reward model, which evaluates the quality of self-explored data. Many existing works derive a single outcome reward based on ground truth (Zelikman et al., 2022; Yuan et al., 2023; Singh et al., 2023) or feedback provided by the environment (Song et al., 2024) at the end of trajectories. While this approach is straightforward, it falls short in handling complex tasks, since an outcome reward model cannot accurately score each step within a long trajectory in intricate scenarios. Also, a trajectory achieving a high final outcome reward does not necessarily indicate that every action taken was optimal; the agent may have completed the task successfully, but some actions could have been inefficient or suboptimal (Uesato et al., 2022).

Therefore, a good process reward model is necessary to provide step-wise evaluations of the agent's actions. Such a model enables the agent to fully understand and learn from the intermediate stages of complex tasks, ultimately improving performance and generalization. The key challenge lies in developing an effective process reward model for self-improvement without relying on extensive human annotations for the step-wise reward. There has been a thread of work focusing on process reward modeling (Uesato et al., 2022; Lightman et al., 2023; Wang et al., 2023; Chen et al., 2024).

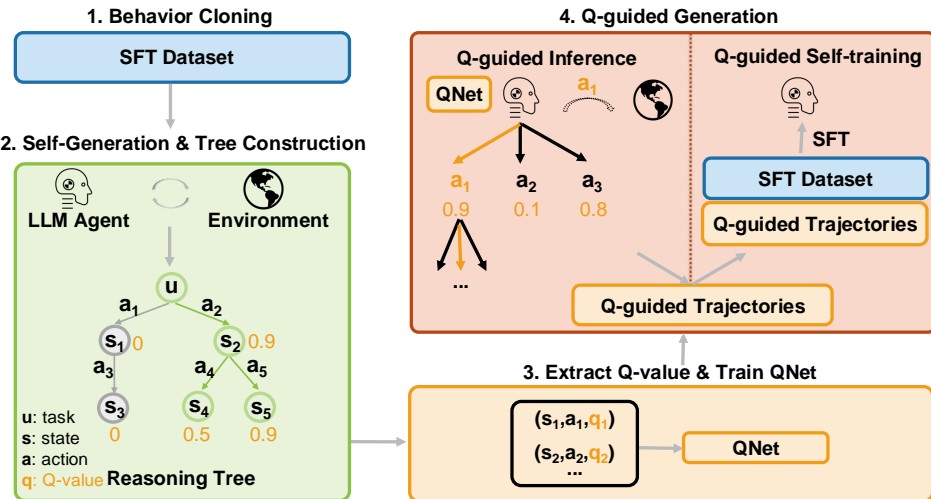

Figure 1: Q*Agent pipeline overview. We revise the original "exploration" teminology in the figure to "generation" to avoid misunderstanding. Q*Agent involves mainly four stages: **1)** Supervised Fine-Tuning on expert data. **2)** Leverage SFT agent to explore the environment and construct a reasoning tree for each task. After construction, estimate the Q-value of each tree node based on Equation 4. **3)** Train QNet on the estimated Q-values. **4)** Use the trained QNet to provide guidance during every exploration step.

However, these methods rely on either costly human-annotation or computationally heavy random rollouts, rendering them inefficient for self-improvement of language model agents.

To address this issue, we propose Q*Agent, a novel approach to provide process guidance with estimated Q value for open language agents. This process reward can be applied to not only boosting self-improvement techniques but also providing direct guidance during inference. As illustrated in Figure 1, we first utilize behavioral cloning to train a base language agent and then ~~do exploration in~~ construct a tree structure ~~to collect~~ collecting a large number of trajectories. With the collected reasoning tree, we use Bellman equation (Bellman & Dreyfus, 2015) to obtain the supervision with state, action, and Q value. Then use the supervision to train a QNet to estimate Q value (Watkins & Dayan, 1992) given any state and action on the reasoning trees. After that, we leverage the trained QNet to collect high quality trajectories in the agent environment. Based on the trained QNet, we propose Q-guided ~~exploration~~ generation to conduct greedy planning in a step-wise manner. We further use the large language models to augment the context to improve the diversity, and design several tree pruning strategies to reduce the redundancy of large searching space.

To summarize, our contribution can be divided into three folds:

**1) Process Reward Modeling with Q-Value Estimation**: We introduce Q*Agent, a novel strategy which leverages estimated Q-values to generate intermediate annotations for language agents, providing effective step-wise guidance for self-improvement.

**2) Q-Guided ~~Exploration~~ Generation Strategy**: We propose a Q-guided ~~exploration~~ generation technique that significantly enhances agent performance by delivering effective process-based guidance during inference, improving decision-making at each step.

**3) Efficient Performance with Limited Supervision**: We mainly evaluate Q*Agent on web navigation tasks, where Q*Agent ~~Our method~~ demonstrates strong performance even when using nearly half the amount of annotated data, highlighting the efficiency and robustness of Q*Agent in scenarios with limited supervision.

## 2 RELATED WORK

### 2.1 LARGE LANGUAGE MODEL AGENT

Large language models have shown impressive performance in complex interactive tasks, such as web navigation (Yao et al., 2022), scientific reasoning (Wang et al., 2022; 2024b), and action planning

in embodied environments (Shridhar et al., 2021). ReAct (Yao et al., 2023) developed a prompting method to shape language models as agents that can reason and act. While several works (Shen et al., 2024; Song et al., 2023) improve agent performance with closed-source LLM controllers, the open-source LLM agents still offer unique advantages like accessibility and customization. FireAct (Chen et al., 2023) and LUMOS (Yin et al., 2024) leverage high-quality data generated by experts and employ teacher-forcing to improve the performance of open-source agents. In line with this, our Q$^\star$Agent is also based on open-source LLMs.

## 2.2 SELF-IMPROVEMENT OF LLM AGENTS

Training model on self-generated data is a promising approach as it circumvents the high cost of collecting expert data. A large number of works (Dou et al., 2024; Zelikman et al., 2022; Yuan et al., 2023; Singh et al., 2023) follow the paradigm of reinforced self-training (Gulcehre et al., 2023), which filters positive self-generated data and performs model training on those filtered positive data. Some other works (Song et al., 2024; Setlur et al., 2024) utilize both positive and negative data to construct preference pairs and update the policy using direct preference optimization (Rafailov et al., 2024). Most of these works rely on the outcome rewards to distinguish between positive and negative trajectories. However, our Q$^\star$Agent can provide process reward signals for intermediate states and actions of a trajectory. Most recently, Wang et al. (2024a) and Zhai et al. (2024) uses step-level guidance for agent inference through training a step-level value model. Putta et al. (2024) applies a hybrid process reward modeling for web navigation tasks by combining Monte Carlo Tree Search (MCTS) rewards with scores generated by large language models to form process rewards. Our method differs from Wang et al. (2024a) and Zhai et al. (2024) in engaging behavioral cloning stage, and differs from Putta et al. (2024) because we do not rely on an external LLM to provide rewards.

## 2.3 PROCESS REWARD MODELING FOR LLM

Existing works have explored various strategies and reasoning policies for process reward modeling. Uesato et al. (2022) and Lightman et al. (2023) utilize human-annotated step-level correctness to train a reward model. while Math-Shepherd (Wang et al., 2023) infers per-step rewards through random rollouts. TS-LLM (Feng et al., 2023) employs an MCTS-based policy and infers per-step rewards using the TD-$\lambda$ (Sutton, 1988) method. V-STaR (Hosseini et al., 2024) and Self-Rewarding (Yuan et al., 2024) leverage the Chain-of-Thought (CoT) reasoning policy, generating final outcome rewards either through multi-iteration LLMs or LLMs' own judgment. ReST-MCTS* (Zhang et al., 2024) uses Monte Carlo tree search (MCTS) with re-inforced self-training to enhance the diversity and performance on general reasoning tasks like maths, science and code. Our approach, focuses more on the agent tasks which require dense interaction with the environment. Also, distinct from these, our method models process rewards using Q-learning. By inferring per-step process rewards through the bellman equation, we effectively capture and optimize the intermediate reasoning steps, enhancing self-improvement capabilities in multi-step reasoning tasks.

## 3 PRELIMINARIES

In this section, we introduce key foundational concepts relevant to Q$^\star$Agent. We begin by discussing Q-learning, which serves as the inspiration for Q$^\star$Agent by extracting Q-values from the reasoning tree. Following that, we will cover the self-improvement techniques which our Q$^\star$Agent aims to provide guidance for.

### 3.1 Q-LEARNING: LONG-TERM VALUE IN DECISION MAKING

Q-learning (Watkins & Dayan, 1992) is a traditional model-free reinforcement learning algorithm, where agents learn a Q-function $Q(s_t, a_t)$ representing the expected future rewards by taking action $a_t$ in state $s_t$ at step $t$. In Q-learning, Q-function is updated iteratively by

$$Q(s_t, a_t) \leftarrow Q(s_t, a_t) + \alpha \left[ R_t + \gamma \max_{a \in \mathcal{A}} Q(s_{t+1}, a) - Q(s_t, a_t) \right], \tag{1}$$

where $\alpha$ is the learning rate, $\gamma$ is the discount factor, $\mathcal{A}$ is the action space and $R_t$ represents the intermediate reward at step $t$. Combining both immediate rewards from the current action and future

---

**Algorithm 1** General Q$^\star$ Agent Pipeline

---

1: **Input:** $\mathcal{D}_{expert} = \{(u_i, a_t^i, o_t^i)_{t=1}^T\}_{i=1}^N$, Policy model $\pi_\theta$, QNet $\mathcal{Q}_\phi$                    ▷ Initialization
2: **Stage 1: Behavior Cloning**
3: Train $\pi_\theta$ on $D_{expert}$ minimizing loss 3
4: **Stage 2: ~~Explore and~~ Construct reasoning trees**
5: **for** $i = 1$ to $N$ **do**                                                                                        ▷ Explore the task $u_i$
6:     Explore on task $u_i$ and construct a reasoning tree $T_i$ with a root node U$_i$
7:     Update Q-values recursively from U$_i$ using Equation 4
8: Collect Q-values from $\{T_i\}_{i=1}^N$ as dataset $\mathcal{D}_Q$
9: **Stage 3: QNet Training**
10: Train QNet $\mathcal{Q}_\phi$ on dataset $\mathcal{D}_Q$
11: **Step 4: Q-guided ~~Exploration~~Generation**
12: Use QNet $\mathcal{Q}_\phi$ to score state-actions at each step                    ▷ Flexible to conduct self-improvement

---

potential rewards from subsequent actions, Q-value can be interpreted as the expected long-term value of taking a specific action in a given state, followed by the optimal policy thereafter. The Bellman Optimality Equation (Bellman & Dreyfus, 2015) of Q-function can be written as

$$Q^\star(s_t, a_t) = R_t + \gamma \max_{a \in \mathcal{A}} Q^\star(s_{t+1}, a). \tag{2}$$

In complex interactive tasks, the agent needs to account not only for immediate rewards but also for the potential long-term effects of its current decisions. This is where the Q-value becomes essential. However, directly adapting RL algorithms to language agents can be sample-inefficient (Jin et al., 2018). This is because the action space in language agent tasks is typically a vast vocabulary, which may lead to an explosion of potential action sequences to be explored. To address this challenge, our approach successfully adapts Q-value extraction to language agent tasks by introducing a reasoning tree, which we will introduce in the next section.

## 3.2 SELF-IMPROVEMENT

Self-improvement is referred to techniques where models leverage self-generated data to improve themselves. Self-training (Altun et al., 2005) is one of the self-improvement techniques that train the model on selected self-generated data. It commonly consists of two stages: `grow` and `improve` Gulcehre et al. (2023). In the first `grow` stage, an augmented dataset $\mathcal{D}_g$ will be created by sampling a set of sequences from the current policy model $\pi_\theta$. The newly generated sequences will be scored by a reward function. Only those sequenced whose score is better than a pre-defined threshold will be retained in $\mathcal{D}'_g$. Then in the second `improve` stage, the current policy model $\pi_\theta$ will be trained on the selected dataset $\mathcal{D}'_g$.

In addition to training-based methods, self-generated data can be leveraged to provide direct inference guidance. In complex agent tasks, inference-time guidance becomes particularly important due to the high cost of collecting expert-annotated data. In our work, we evaluate whether Q$^\star$ Agent can enhance the performance of self-improvement techniques through two setups: the first focuses on providing direct guidance during inference, while the second involves Q-guided self-training. In the latter experimental setup, the self-training data is generated under the guidance of QNet. We will provide further details in the following section.

## 4 METHODOLOGY

In this section, we will follow the order of Q$^\star$ Agent training pipeline and introduce each critical component step by step. The overall pipeline is stated in Figure 1 and Algorithm 1. First, we will describe the initial stage of behavior cloning. Then, we will explain how the reasoning tree is constructed during the second explore stage and how we utilize it to extract Q-values. Finally, we will detail how the Q-network (QNet) is employed to guide the agent's generation process and to boost self-improvement techniques.

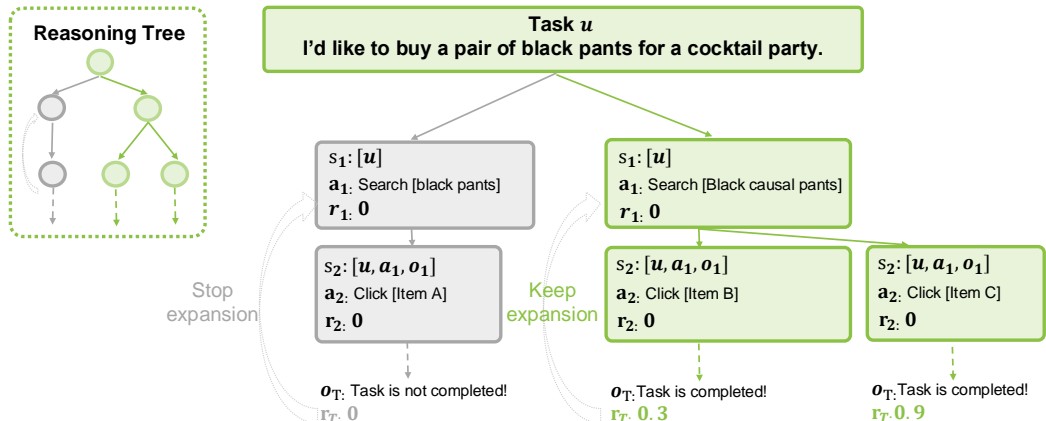

Figure 2: Note: Updated figure aligned with WebShop. Illustrative example of constructing a reasoning tree. Grey nodes represent the branches with a zero outcome reward. Once the leaf node with a zero outcome reward is detected, a `Stop expansion` signal will be sent back to the first unexpanded node on the branch. Green nodes are on branches where zero outcome reward is not detected and can keep expanding.

## 4.1 BEHAVIORAL CLONING

Behavior cloning provides a strong initial foundation for language agents by supervised fine-tuning on expert trajectories. Formally, the first stage of Q$^\star$Agent is to supervised fine-tune our language agent, denoted as the policy $\pi$, on a set of annotated samples $\mathcal{D}_{expert}$. We use ReAct (Yao et al., 2023)-style data for supervised fine-tuning, which additionally generates Chain-of-Thought (CoT) (Wei et al., 2022) reasoning paths before executing each action. We will use $a$ to denote the complete ReAct-style response generated by $\pi$ for simplicity.

Formally, given a dataset $\mathcal{D}_{expert} = \{(u_i, a_t^i, o_t^i)_{t=1}^T\}_{i=1}^N$, where $u_i$ represents the task description, $T$ is the trajectory length, $N$ is the number of trajectories in expert dataset, $o_t^i$ is the environment observation after taking action $a_t^i$ at step $t$, we optimize the policy $\pi$ by minimizing the ~~cross-entropy loss~~ negative log-likelihood loss:

$$\mathcal{L}(\theta) = -\sum_i \sum_t \log \pi_\theta(a_t^i \mid u_i, a_{<t}^i, o_{<t}^i), \tag{3}$$

where $\theta$ denotes the parameters of the policy model $\pi_\theta(a_t|u, h_t)$, which outputs the probability of action $a$ given task description $u$ and historical interactions $h_t = \{a_{<t}, o_{<t}\}$.

## 4.2 CONSTRUCTING A REASONING TREE

The supervised fine-tuned agents can explore the environment and collect a large amount of trajectories. However, due to the extremely large action space of language agents, directly sampling trajectories without any guidance may lead to low ~~exploration~~generation efficiency. To address this issue, we propose to construct a reasoning tree during self-~~exploration~~generation to enhance ~~exploration~~generation to enhance search efficiency.

### 4.2.1 TREE STRUCTURE

For a trajectory, we take the task description as the root node and formalize it into a branch, where each step's state, action, and related information form a node. For all trajectories of a task, they can be seen as different branches originating from the same root node.

Specifically, a **TreeNode** $N$ in a Reasoning Tree is defined as follows:

**State** ($s_t$): Represents the accumulated historical context from the initiation of the process up to the current time step $t$, encapsulating all preceding reasoning paths and actions. Formally, the state at time $t$ is given by

$$s_t = \{u, a_1, o_1, \ldots, a_{t-1}, o_{t-1}\},$$

including the initial task description $u$ and interactive history at step $t$.

**Action** ($a_t$): denotes the specific operation performed at the current node, which affects the subsequent state. The action is selected by the policy language agent $\pi$ and is conditioned on the current state and reasoning path.

**Reward** ($r_t$): the immediate feedback received from environment after performing action $a_t$. In most language agent tasks, the immediate rewards from environments are set to zero or very sparse. For example, WebShop (Yao et al., 2022) only provides a final reward from 0 to 1 at the end of trajectories.

**Children** ($\mathcal{C}$): is represented by a list containing nodes explored at the next step.

**Q-value** ($q$): represents the expected total future reward achievable starting from the current state $s_t$, taking action $a_t$. The Q-values are updated once a reasoning tree is constructed. We will introduce how we extract Q-values in the following section.

### 4.2.2 TREE CONSTRUCTION

With each step in a trajectory formalized as a TreeNode, the entire trajectory is a branch within a reasoning tree. To explicitly construct a reasoning tree that captures potential ~~exploration~~generations from the root node (i.e., the initial task), exploring new trajectories can be viewed as expanding new branches from the existing TreeNodes. For any non-leaf tree node, effective ~~exploration~~generation can be achieved by: **1)** directly exploring and adding new child nodes that differ from the existing ones. **2)** For each branch that reaches a leaf node, we assess its quality based on the final reward. If the branch yields a zero reward, we stop ~~exploration~~generation on that branch's nodes, thereby reducing ineffective ~~exploration~~generation.

**Tree Pruning.** In practice, we have found that the average depths of tree searching for agent tasks are large. Building a reasoning tree and expanding every potential tree nodes may lead to heavy cost to the trajectory ~~exploration~~generation. To address this, we propose several strategies to reduce the computational burden during tree construction. We employ pre-pruning techniques to lower the ~~exploration~~generation costs when constructing a reasoning tree for each task. First, we limit the expansion of tree nodes to the early stages of a trajectory (e.g., the first three to five steps, depending on the environment's complexity, with details provided in Appendix A.1).

Next, when a branch leads to a zero-outcome reward at its leaf node, we propagate a `Stop expansion` signal from the leaf node back to the earliest unexpanded intermediate node on that branch. This helps prioritize the ~~exploration~~generation of optimal trajectories given a limited ~~exploration~~generation budget. This construction process is illustrated in Figure 2. With a set of reasoning trees, we aim to gather effective step-wise signals for training an effective process reward model. Since most language agent tasks only return an outcome reward at the end of the trajectory, which is stored at the leaf nodes of the reasoning tree, we need to develop methods to leverage these outcome rewards to generate effective intermediate signals.

**Extracting Q-values.** After constructing a reasoning tree, with the final outcome rewards stored in leaf node rewards, we estimate the Q-values for each intermediate nodes leveraging

$$Q(s_t, a_t) = r_t + \gamma \max_{a_{t+1} \sim \mathcal{C}_t} [Q(s_{t+1}, a_{t+1})], \tag{4}$$

where $\gamma$ is the discount factor, $s_{t+1}$ is the new state after action $a_t$, $\mathcal{C}_t$ is the children set containing nodes explored at the next step, and the expectation is over actions $a_{t+1}$ drawn from the policy $\pi$. We provide the pseudocode of tree construction and Q-value estimation on the reasoning trees in Appendix A.4.

### 4.3 QNET TRAINING

Inspired by the value function representing the expected long-term value in Q-learning (Watkins & Dayan, 1992), we extract Q-values for each nodes on reasoning trees using Equation 4. For each node $N = (s, a, q, ..)$ in the collected reasoning trees, we can extract a supervised dataset $D_Q = \{(s, a, q)\}$ to train Q-network (QNet). The model architecture of QNet is introduced in Appendix A.2

**Training Objective:** Given each reasoning tree with $n$ nodes: **Tree** $= (N_1, N_2, \ldots, N_n)$, we train the QNet $\mathcal{Q}_\phi$ by minimizing the Mean Squared Error (MSE) loss between the predicted Q-values $\hat{q}_t$

Table 1: Performance overview of all methods. The table is divided into three sections: the first presents the results of closed-source agents, the second includes training-based methods, and the third shows inference algorithm results. Our results are averaged rewards on the test set with 200 instructions. In each section, the best result is highlighted in **bold**, while the second-best result is underlined.

| Method | WebShop |
|---|---|
| GPT-4 | 63.2 |
| GPT-3.5-Turbo | 62.4 |
| Reflexion (Shinn et al., 2023)[1] | 64.2 |
| LATS (Zhou et al., 2024)[1] | 75.9 |
| Llama-2-7B-Chat | 17.9 |
| Llama-2-7B-Chat + SFT | 63.1 |
| Llama-2-7B-Chat + RFT | 63.6 |
| Llama-2-7B-Chat+Q$^\star$Agent-ST | 66.4 |
| Llama-2-7B-Chat + PPO | 64.2 |
| Llama-2-7B-Chat + ETO | **67.4** |
| Llama-2-7B-Chat + Best-of-N | 65.3 |
| Llama-2-7B-Chat + Best-of-N-aug | 68.4 |
| Llama-2-7B-Chat + Q$^\star$Agent-I | 65.5 |
| Llama-2-7B-Chat + Q$^\star$Agent-I-aug | **72.6** |

and the provided Q-value $q$ at each time step:

$$\mathcal{L}(\phi) = \frac{1}{n} \sum_{t=1}^{n} \left( \hat{q}_t - q_t \right)^2 . \tag{5}$$

By minimizing this loss, we encourage the QNet to produce consistent Q-value estimations across the sequence that align with the target Q-value $q$. This training objective emphasizes accurate Q-value predictions at each token, reinforcing the model's ability to assess the long-term value of actions throughout the trajectory.

### 4.4 Q-GUIDED ~~EXPLORATION~~ GENERATION

The effectiveness of a good process reward model can be represented by whether it can lead to better agent self-improvement. Therefore, we conduct Q-guided ~~exploration~~ generation for self-improvement to evaluate the effectiveness of Q$^\star$Agent. Q-guided ~~exploration~~ generation enables agents to generate each step under the guidance of QNet. At each step, agents sample several actions and the one with the highest Q-value is executed by the agent. We provide a more detailed algorithm of Q-guided ~~exploration~~ generation in Appendix A.3.

**Perturbation augmented ~~exploration~~ generation.** To augment the samples actions at each step, we also introduce augmenting action diversity with perturbation during this stage, which is realized by prompting LLM to paraphrase the task description. This utilization of perturbation enables us to inject more variability into the prompts that guide action selection, substantially enriching the range and relevance of possible actions. Such enhanced prompts help prepare the model to handle more diverse and unforeseen situations effectively. We provide our implementation details and examples in Appendix A.5.

In this section, we introduce Q$^\star$Agent, a strategy that leverages Q-value estimation for process reward modeling, providing step-wise guidance for language agents. Additionally, we propose a Q-guided ~~exploration~~ generation strategy that enhances the agent's decision-making by using Q-values to drive more effective ~~exploration~~ generation during inference.

## 5 EXPERIMENT

---

[1]These results are adopted from Zhou et al. (2024).

In this section, we aim to evaluate the effectiveness of Q⋆Agent for solving complex agent tasks in the following aspects: 1) Whether Q⋆Agent can aid better self-improvement by providing inference-time guidance or by selecting better data for self-training; 2) Qualitative analysis on the Q-guided agent generation to see whether Q⋆Agent can provide effective guidance for each step; 3) Ablation study on different variants of process rewards extracted from reasoning trees.

## 5.1 SETUP

**Dataset.** We assess the ability of Q⋆Agent on WebShop (Yao et al., 2022), a realistic web navigation benchmark, where an agent is required to explore various types of web pages, perform different actions, and ultimately locate, customize, and purchase an item given a text instruction detailing a product. Following the setup of ETO (Song et al., 2024), we use a training data consisting of 1938 trajectories for behavior cloning and 200 instructions for testing. The evaluation metric is the reward averaged on 200 instructions in the test set. During sampling process, the environment will give termination signal after certain action "Click" or achieve the maximum steps set in advance. Specifically, we set the maximum as 5 for WebShop during self-generation and Q-guided generation.

**Backbone.** In our work, we mainly use Llama-2-7B-Chat as base policy model and QNet backbone. The detailed hyper-parameters for training and model architectures can be found in Appendix A.1.

To fully assess the effectiveness of Q⋆Agent, we develop several variants for Q⋆Agent, denoted as Q⋆Agent-I, Q⋆Agent-aug and Q⋆Agent-ST respectively. 1) **Q⋆Agent-I**: Q⋆Agent can provide direct step-wise guidance for action generation during inference. We can refer to this variant of Q⋆Agent as Q⋆Agent-I. 2) **Q⋆Agent-I-aug**: Based on Q⋆Agent-I, we use GPT-3.5-Turbo to do perturbation introduced in Section 4.4 to augment task descriptions during Q-guided ~~exploration~~generation, which is denoted as Q⋆Agent-I-aug. 3) **Q⋆Agent-ST**: This Q⋆Agent leverages QNet to select data for self-training by combining SFT data with self-generated data where multiple actions are sampled at each step and the one with the highest Q-value is selected.

**Baselines**. 1) **SFT** (Chen et al., 2023) is the base agent after supervised fine-tuning on the expert data. 2) **RFT** (Rejection sampling Fine-Tuning) (Yuan et al., 2023) is a self-improvement baseline which is trained on the merged data consisting of successful trajectories sampled and expert data. 3) **ETO** (Song et al., 2024) is a self-improvement baseline which updates policy via constructing trajectory-level preference pairs and conducting DPO. 4) **PPO** (Proximal Policy Optimization) (Schulman et al., 2017): a reinforcement learning baseline which directly trains the base agents to optimize the final rewards. 5) **Best-of-N** samples N trajectories for each task and selects the one with highest outcome reward. For fairer comparison among inference algorithms, we also develop a variant of Best-of-N which also adopts perturbation introduced in Section 4.4 denoted as **Best-of-N-aug** for a fair comparison with Q⋆Agent-I-aug. ~~N is set to 6 in Table 1 and Table 2~~. N is set to 10 in Table 1 and 6 in Table 2. All inference algorithms in the tables are under the same search budget. 6) **Closed-source agents** including GPT-3.5-Turbo and GPT-4 with ReAct prompting (Yao et al., 2023), and other methods depending on the emergent properties of self-reflection and planning from large proprietary models, such as Reflexion (Shinn et al., 2023) and LATS (Zhou et al., 2024).

## 5.2 SELF-IMPROVEMENT PERFORMANCE

In this section, we compare the performance of our Q⋆Agent for self-improvement with all the baselines. Results are summarized in Table 1. We evaluate all algorithms using one-shot evaluation. From Table 1, we can observe that Q⋆Agent-I-aug achieves the highest score among all the training-based and inference-based algorithms, with comparable performance to the best agent depending on proprietary models.

### 5.2.1 SELF-TRAINING

~~Table 2~~ Table 1 is organized into three sections: the first section presents the results of closed-source agents, the second covers training-based approaches, including self-training methods (RFT and Q⋆Agent-ST), reinforcement learning (RL), and DPO-based optimization, and the third section highlights inference algorithms. Q⋆Agent-ST achieves the second-best result among the training-based methods and the best result among the self-training methods.

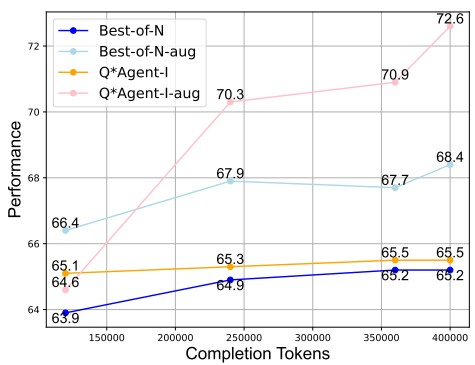
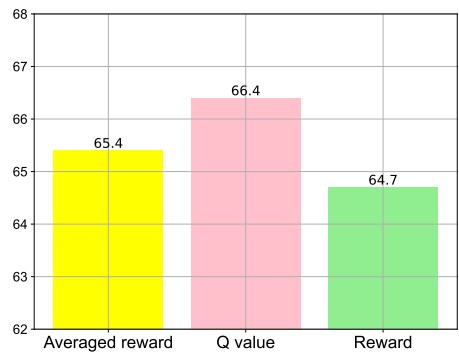

(a) Comparison on inference performance.  (b) Comparison of different process rewards.

Figure 3: Left: Inference algorithms comparison with varying completion tokens. Right: Process rewards comparison. `Q value` is adopted in Q⋆Agent. The evaluation metrics in two figures are both averaged rewards on test instructions.

Table 2: ~~Performance~~Average reward comparison on WebShop with 1000 annotated trajectories for behavior cloning. The best result is **bolded**, and the second-best result is underlined.

| Method | WebShop | WebShop-1000 |
|---|---|---|
| Llama-2-7B-Chat + SFT | 63.1 | 21.7 |
| Llama-2-7B-Chat + RFT | 63.6 | 61.4 |
| Llama-2-7B-Chat + ETO | 67.4 | 66.7 |
| Llama-2-7B-Chat + Best-of-N | 64.9 | 24.5 |
| Llama-2-7B-Chat + Best-of-N-aug | 67.9 | 47.1 |
| Llama-2-7B-Chat + Q⋆Agent-I | 65.3 | **68.2** |
| Llama-2-7B-Chat + Q⋆Agent-I-aug | **70.3** | 67.3 |

Comparing Q⋆Agent-ST and RFT, we find that Q⋆Agent-ST demonstrates better performance. The key difference between the two methods lies in how the self-training data is selected: Q⋆Agent-ST uses Q-guided ~~exploration~~generation to choose data in a step-wise manner, while RFT selects successful trajectories based on the environment's final outcome reward. Therefore, the improved performance of Q⋆Agent-ST may be led by better data selection through Q-guided ~~exploration~~generation. Additionally, a concurrent work by Zhai et al. (2024) also conducted experiments on WebShop following a similar ETO setup and SFT dataset. They used Llama-3.1-8b-instruct as their base agent and achieved a final reward of 60.

### 5.2.2 INFERENCE-TIME PERFORMANCE

We compare all the inference baselines under different search budgets. As shown in the Figure 3a, increasing the number of completed / generated tokens (the number of explored trajectories) will improve the performance in all inference methods. We can observe that Q⋆Agent-I-aug achieves the best performance under almost all the search budgets. Another notable observation is that compared with Best-of-N-aug under 400k tokens, Q⋆Agent-I-aug outperforms the highest score of Best-of-N-aug(68.4) under 400k tokens with only about half of search budgets(70.3) under 240k tokens. Also, as the completed tokens approach 360K, other approaches begin to flatten, while Q⋆Agent still improves from 360K tokens to 400K tokens. This indicates that our approach is a more effective way to scale up the compute to inference-time self-improvement.

### 5.3 FEWER ANNOTATIONS

In many real-world applications, collecting large amounts of expert-annotated data is both time-consuming and costly. To evaluate the effectiveness of our approach under such constraints, we

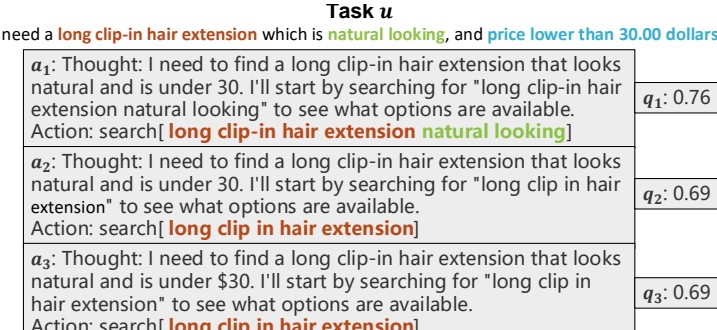

Figure 4: A case study on WebShop.

designed this setup with fewer annotations to test its robustness in this section. We extract 1000 trajectories as a subset from the original 1938 trajectories. Under this setup, all baselines can only conduct behavior cloning with access to the SFT dataset of 1k trajectories. After that, baselines like RFT, ETO and Q*Agent which involve ~~exploration~~generation can explore on 1938 tasks. The performance comparison is listed in Table 2. We can observe that Q*Agent-I-aug outperforms other methods on both the full WebShop dataset and the WebShop-1000 subset. This highlights the robustness of our method, especially in scenarios with scarce expert data. While other methods like RFT and SFT show a significant drop in performance, Q*Agent-I-aug remains effective, proving the advantage of Q-guided ~~exploration~~generation for data selection even in annotation-limited environments.

### 5.4 QUALITATIVE ANALYSIS ON GENERATED RESPONSES

In addition to quantitative experiments, we also aim to assess whether the Q-value can correctly evaluate the quality of intermediate actions. Therefore, we visualized a case in the WebShop environment, where the first step of the trajectory typically involves the agent searching relevant keywords into a webpage based on the instructions. As shown in Figure 4, the original task specifies three attributes for the item, each highlighted in a different color. Below, the agent samples three actions. The last two actions capture only one attribute during the search, while $a_1$ captures two attributes. As expected, the Q-value for $a_1$ should be higher. QNet scores these three actions, and indeed, action 1 receives the highest Q-value, aligning with our direct observations.

### 5.5 ABLATION STUDY OF PROCESS REWARD MODELING

Since process reward modeling is an important module in our framework, we ablate on how different choices of process reward can affect the performance. We mainly experiment with three approaches of constructing process rewards for each intermediate nodes on the reasoning trees: `Q value`(ours) is to estimate Q-value for each state-action pair (i.e. each tree node except for root node) using Equation 4; `Averaged reward` computes the averaged children rewards; `Reward` directly treats the final outcome reward as the process reward for each step. We train three different process reward models guiding trajectory generation for self-training. Self-training results are in Figure 3b. From Figure 3b, we can observe that `Q value` utilized by our Q*Agent yields the best performance, while the one using `Averaged reward` is slightly better than the one directly using `Reward`, indicating the effectiveness of using `Q value` to model process reward.

## 6 CONCLUSION

In this paper, we introduce Q*Agent, a novel approach that enhances the self-improvement capabilities of open-source language models by integrating Q value-based process guidance. By modeling the Q value at each intermediate step during planning, our method offers step-wise feedback that surpasses the limitations of outcome-based reward models, particularly in complex, long-horizon tasks.

Through extensive experiments, we have demonstrated that Q*Agent significantly improves the model's ability to generate high-quality trajectories, ultimately leading to better performance in both self-improvement and inference tasks. Moreover, our method demonstrates strong performance even

in scenarios with limited annotated data, highlighting the efficiency and robustness of our $Q^\star$ Agent. This work paves the way for more efficient and scalable self-improvement techniques in language models, enabling them to tackle complex tasks with reduced reliance on human annotations.

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

# A APPENDIX

## A.1 EXPERIMENTAL DETAILS

### A.1.1 DATASETS

We follow the setup of ETO (Song et al., 2024) to use the classical WebShop for agent training and evaluation. WebShop is an online shopping environment. The available action types for agents include *search[keywords]* and *click[value]*. The agent is instructed to complete the task with ReActYao et al. (2023)-style response. The instruction is specified in Figure 5

### A.1.2 HYPER-PARAMETERS

We summarize the hyper-parameters used across both all stages of Q\*Agent in this section. The hyper-parameters leveraged in behavior cloning and self-training is in Table 3. Training QNet shares all the same hyperparameters, except that the number of training epochs is set to 2.

## A.2 QNET

**Model Architecture:** Our QNet is designed by sharing the backbone of the Large Language Model (LLM) and appending a value head to predict Q-values. Specifically, we utilize a pre-trained LLM, denoted as $\text{LLM}_\theta$, which serves as the foundational model for encoding input sequences. The value head is a Multi-Layer Perceptron (MLP) that takes the hidden states from the LLM and outputs scalar Q-value predictions.

Formally, given an input sequence of tokens $\mathbf{x} = (x_1, x_2, \ldots, x_n)$, the LLM produces hidden states $\mathbf{h} = (h_1, h_2, \ldots, h_n)$:

$$\mathbf{h} = \text{LLM}_\theta(\mathbf{x}), \tag{6}$$

where $h_t \in \mathbb{R}^d$ represents the hidden state at time step $t$, and $d$ is the hidden size of the LLM.

The value head $\text{MLP}_\phi$ processes each hidden state $h_t$ to predict the corresponding Q-value $\hat{q}_t$:

$$\hat{q}_t = \text{MLP}_\phi(h_t), \tag{7}$$

where $\hat{q}_t \in \mathbb{R}$ is the predicted Q-value at time step $t$, and $\phi$ denotes the parameters of the MLP.

The MLP consists of multiple layers with ReLU activations, culminating in a linear layer that outputs a scalar Q-value. This design allows the model to capture complex patterns in the hidden representations and map them to accurate Q-value estimates.

**Training Objective:** Given an explored trajectory $\mathbf{x} = (x_1, x_2, \ldots, x_n)$ with an associated target Q-value $q$, we train the QNet by minimizing the Mean Squared Error (MSE) loss between the predicted Q-values $\hat{q}_t$ and the provided Q-value $q$ at each time step:

$$\mathcal{L}(\theta, \phi) = \frac{1}{n} \sum_{t=1}^{n} (\hat{q}_t - q)^2. \tag{8}$$

By minimizing this loss, we encourage the QNet to produce consistent Q-value estimations across the sequence that align with the target Q-value $q$. This training objective emphasizes accurate Q-value predictions at each token, reinforcing the model's ability to assess the long-term value of actions throughout the trajectory.

**Implementation Details:** In practice, we implement the value head as an MLP with two hidden layers of size 1024 and ReLU activation functions:

$$\text{MLP}_\phi(h_t) = \text{Linear}_3(\text{ReLU}(\text{Linear}_2(\text{ReLU}(\text{Linear}_1(h_t))))), \qquad (9)$$

$$\text{where} \quad \text{Linear}_1 : \mathbb{R}^d \to \mathbb{R}^{1024}, \qquad (10)$$

$$\text{Linear}_2 : \mathbb{R}^{1024} \to \mathbb{R}^{1024}, \qquad (11)$$

$$\text{Linear}_3 : \mathbb{R}^{1024} \to \mathbb{R}. \qquad (12)$$

The entire model, including the LLM and the value head, operates in bfloat16 precision to optimize memory usage without sacrificing performance. The LLM backbone remains frozen or fine-tuned depending on the specific experimental setup, allowing us to leverage pre-trained language representations while focusing on learning accurate Q-value predictions through the value head.

By integrating the value head with the LLM, our QNet effectively combines language understanding with reinforcement learning principles, enabling the agent to make informed decisions based on both linguistic context and estimated future rewards.

### A.3 Q-GUIDED ~~EXPLORATION~~GENERATION

In this section, we present the pseudocode of Q-guided ~~exploration~~generation in Algorithm 2, which is a critical component of our framework.

### A.4 PSEUDOCODE OF REASONING TREE CONSTRUCTION AND Q-VALUE DISTILLATION.

In this section, we provide the pseudocode of constructing a reasoning tree in stage 2 in Algorithm 3 and and how we distill the Q-value from a reasoning tree in Algorithm 4.

### A.5 PERTURBATION AUGMENTED GENERATION

---

**Algorithm 2** Q-guided Exploration

---

1: **Input**: A LLM agent $\pi_\theta$, a given task description $u$, an action set $\mathcal{A}_t$ containing $M$ candidates at step $t$, a trained QNet $\mathcal{Q}_\phi$, sampled trajectory number $N$, max trajectory length $L$
2: traj_candidates = [ ]
3: **for** $i = 1$ to $N$ **do**
4:     Initialize state $s_i \leftarrow [u]$
5:     **for** $t = 1$ to $L$ **do**
6:         Collect a set of action candidates $\mathcal{A}_t \leftarrow$ Sample $a \sim \pi_\theta(a \mid s_i)$ for $M$ times
7:         $a_t \leftarrow \text{argmax}_{a \sim \mathcal{A}_t} Q_\phi(s_i, a)$        ▷ Select the best action with max Q-value
8:         Take action $a_t$, and receive new observation $o_t$ from environment
9:         $s_i \leftarrow s_i + [a_t, o_t]$       ▷ Update state with executed action and new observation
10:         **if** $s_i$ is the final state **then**
11:             **break**               ▷ Exit loop if stop condition is met
12:     traj_candidates.append($s_i$)
13: Select the best trajectory $s$ with best final reward $s$.reward from traj_candidates

---

We use GPT-3.5-turbo to perturb the task descriptions using the prompt *" Paraphrase the text: task description "*. We also provide an illustrative example on a WebShop task in Figure 6.

**Algorithm 3** Constructing a Reasoning Tree

1: **Input**: A LLM agent $\pi_\theta$, a given task description $u$, a trajectory $\tau_0$ from the training set $\mathcal{D}_{expert}$ on task $u$, max exploration depth $D$, max exploration width $W$
2: Initialize a root node $U$ with state $s \leftarrow u$, depth $t \leftarrow 0$, reward $r \leftarrow 0$, action $\leftarrow$ *null*, children set $\mathcal{C} \leftarrow \{\}$
3: Initialize the reasoning tree $\mathcal{T}$ with $U$
4: The expansion node queue $E \leftarrow [u]$
5: **while** $E$ is not empty **do**
6:     Get a node $N \leftarrow E$.pop with state $N.s$, action $N.a$, reward $N.r$, children set $\mathcal{C}$ at step $N.t$
7:     **if** the number of children in $N.\mathcal{C} < W$ and $N.t <= D$ **then**
8:         Sample a new trajectory $\tau$ based on state $N.s$
9:         Get a new branch $b$ constructed on $\tau$ and merge $b$ in node $N.\mathcal{C}$
10:         **if** $\tau$ achieves a non-zero final reward **then**
11:             Push all the nodes on $b$ with $N.t <=$ depth $t <= D$ into $E$
12: Construct a branch $b$ with $\tau_0$ and merge in $U.\mathcal{C}$
13: Push all the nodes on $b$ with depth $t$ and $t <= D$ into $E$
14: **Repeat** Function in Line 5-12
15: **return** the reason tree $\mathcal{T}$

---

**Algorithm 4** Q-value Estimation

1: **Input**: A reasoning tree $\mathcal{T}$ with a root node $U$, discount factor $\gamma$
2: **procedure** UPDATE_Q_VALUES($N$)
3:     **if** $N.\mathcal{C} = \emptyset$ **then**            ▷ Check if N is a leaf node
4:         return            ▷ Leaf nodes do not update
5:     **for** node $N_{\text{child}}$ in $N.\mathcal{C}$ **do**
6:         UPDATE_Q_VALUES($N_{\text{child}}$)     ▷ Recursively update child nodes first
7:     $N.q = N.r + \gamma \max_{N_{\text{child}} \sim N.\mathcal{C}}(N_{\text{child}}.q)$   ▷ Update Q-value after all children are updated
8: UPDATE_Q_VALUES($U$)          ▷ Start the update process from the root
9: $Q_{\min} = \min_{N \in \mathcal{T}}(N.q)$
10: $Q_{\max} = \max_{N \in \mathcal{T}}(N.q)$
11: **for** node $N$ in $\mathcal{T}$ **do**
12:     $N.q = \frac{N.q - Q_{\min}}{Q_{\max} - Q_{\min}}$        ▷ Apply min-max normalization
    **return** the reasoning tree $\mathcal{T}$ with estimated Q-value of each node

---

Table 3: Training hyperparameters used in Behavior Cloning and Self-Training.

| Hyperparameter | Value |
|---|:---:|
| Batch size | 64 |
| Number of training epochs | 3 |
| Weight decay | 0.0 |
| Warmup ratio | 0.03 |
| Learning rate | 1e-5 |
| LR scheduler type | Cosine |
| Logging steps | 5 |
| Model max length | 4096 |
| Discount factor $\gamma$ | 0.9 |
| Maximum expansion depth $D$ on WebShop | 3 |
| Action candidate set size $M$ at each step in Q*Agent-ST | 3 |
| Action candidate set size $M$ at each step in Q*Agent-I | 2 |
| Action candidate set size $M$ at each step in Q*Agent-I-aug | 2 |
| Sampled trajectory number $N$ for each task in Q*Agent-ST | 1 |
| Exploration temperature | 0.7 |

**WebShop Instruction**

You are web shopping.
I will give you instructions about what to do.
You have to follow the instructions.
Every round I will give you an observation and a
list of available actions, you have to respond an
action based on the state and instruction.
You can use search action if search is available.
You can click one of the buttons in clickables.
An action should be of the following structure:
**search[keywords]**
**click[value]**
If the action is not valid, perform nothing.
Keywords in search are up to you, but the value
in click must be a value in the list of available
actions.
Remember that your keywords in search should
be carefully designed.
Your response should use the following format:
**Thought: I think ...**
**Action: click[something]**

Figure 5: The instruction prompt provided to language agent on WebShop.

**Perturbation**

Paraphrase the task: I need a long lasting 6.76 fl
oz bottle of l'eau d'issey, and price lower than
100.00 dollars

I'm looking for a 6.76 fl oz bottle of L'eau D'Issey
that lasts a long time and costs less than $100.

I need a durable 6.76 ounce bottle of L'eau
D'Issey, with a price under $100.

I am searching for a long-lasting 6.76 fluid ounce
L'eau D'Issey perfume for less than 100 dollars.

Figure 6: An illustrative example on task perturbation.

