# OpenReview forum: "Q* Agent: Optimizing Language Agents with Q-Guided Exploration"
_ICLR.cc/2025/Conference — Submitted to ICLR 2025_

### Official Review · Reviewer_gDbK · 2024-10-26

**Soundness:** 3
**Presentation:** 3
**Contribution:** 2
**Rating:** 6
**Confidence:** 3

**Summary:**

The authors propose an LLM agent algorithm based on Q-value-guided training and inference-time decision-making to tackle complex interactive (decision-making) tasks. The primary motivation is to provide more efficient feedback to the agent through step-wise rewards. By integrating the LLM agent within an MCTS-like framework, the approach optimizes the LLM agent, addressing the issue of sparse rewards that arise from solely using trajectory rewards in existing algorithms.

**Strengths:**

The main contribution of this paper lies in its ability to construct step-wise rewards for each step using an MCTS-like approach, based solely on the final reward. These rewards are then used to train a Q-value network to support decision-making at inference time.
In the overall algorithm design, after conducting SFT, the authors make a few adjustments to the MCTS. They utilize a “tree pruning” approach to reduce the search space and train a Q-value network based on calculated Q-values. This allows the Q-value network to guide the agent's decision-making at inference time by selecting actions that maximize the Q-value.

**Weaknesses:**

+ **Algorithm Framework**: The core framework of this paper is derived from the MCTS, yet the authors did not evaluate or compare its relationship to MCTS.
  + **Known Environment Assumption**: Unlike tasks such as mathematical reasoning, the interactive tasks in this paper require constructing a reasoning tree using the environment to achieve state transitions (i.e., generating the next state based on the current state and action). However, the authors did not discuss this assumption.
  + **Tree Pruning**: The main improvement of this work over MCTS lies in tree pruning, considering the vast action space for LLM as a generative model. If a simulation fails to get a positive reward, Q* Agent discards the node that attempted to expand. This approach essentially explores a few trajectories that can reach the final goal within an enormous decision space and then performs Q-value extraction. However, for complex tasks that may require dozens of steps to reach the final goal (getting a positive final reward), the probability of finding a successful trajectory is exceedingly low, severely limiting the algorithm's applicability. The authors even restrict expansion to only the first three to five steps, making the algorithm suitable only for tasks requiring exploration at the very beginning of each episode.
  + **Task-specific Q-value Network**: This method requires training a task-specific Q-value network for each task and using the corresponding network during decision-making, limiting the algorithm's generalization capability, which is a core advantage of LLMs (otherwise, a task-specific agent could be directly trained with RL). Using RL to train the LLM with this Q-function could mitigate this issue.
  + **MDP Design**: The state definition includes the entire interaction history. When the interaction sequence becomes lengthy, this introduces a large number of tokens, further constraining the algorithm's performance on complex tasks.
+ **Experimental Results**: Experimentally, the authors only compares the proposed method with some simple baselines on the WebShop benchmark. While it achieves some performance improvement, the improvement is relatively small. Other experimental results do support the effectiveness of the proposed Q-guided decision-making. However, no ablation studies were conducted on techniques like tree pruning, making the experiments insufficiently comprehensive.
  + **Experimental Setup**: Only WebShop is selected as the experimental platform, limiting the results' credibility. The authors also do not show variance or the number of test episodes, making it difficult to assess the impact of randomness on the results, especially given the small performance improvement.
  + **Limited Experimental Content**: It is recommended that the authors conduct ablation studies on techniques in the tree construction phase, particularly on stop expansion and the "early stage" length in tree pruning.

**Questions:**

Please refer to the weaknesses.

---

> ### Author Response · Authors · 2024-11-22
> **Author Response to Reviewer gDbK**
>
> We sincerely thank the reviewer for spending time reading our paper and giving valuable feedback. We address your concerns below.
>
> >  Comparison with MCTS
>
> We would like to argue that compared with the widely-used MCTS, our method shows difference in the following perspectives:
>
> 1. The reward in the MCTS does not take the depth of the trajectory into account, while Q value represents a discounted value for the future outcome, which is better at modeling long-term value.
> 2. MCTS is a model-based setup, which requires the environment to output the states so that they can leverage to do backtracking in their framework during the sampling process, e.g., in Agent Q and MCTS. Our method, which adopts a Q-learning framework, is a model-free setup which does not necessarily need to know the function of how the environment outputs the future states. The sampling process is purely guided by feeding historical states and current action to the QNet.
>
> >  Known Environment Assumption
>
> We would like to clarify that our approach does not rely on constructing a reasoning tree to achieve state transitions. Instead, the reasoning tree structure naturally arises from the characteristics of interactive tasks.
>
> >  Not enough exploration depth
>
> Thanks for the comments. We respectfully argue that using 3~5 steps as the exploration steps for expansion are already enough for Webshop. Specifically, we do additional ablation studies by sampling 150 instructions and construct reasoning trees with different exploration depths of 3, 5, and 7. The average trajectory rewards are in the table. We can observe that current exploration depth 3 maintains a relatively reasonable averaged trajectory quality. For tasks with longer trajectory length, our method is flexible enough to be adapted to larger exploration depth.
>
> | depth | 3    | 5    | 7    |
>  | ------ | ---  | ---  | ---- |
> | Reward | 52.1 | 52.6 | 47.3 |
>
>
> >  Task-specific Q-value Network
>
>  Thanks for the comment. We would like to respectfully argue that our current method treats the QNet as an external module making it more flexible. Also, our current results show, because our QNet is trained on the trajectories collected from the training set of WebShop, and we still observe the performance gain on the test set of WebShop. This actually indicates the generalizability of our proposed method because there are some unseen item types in the test set that have not shown up in the training set, showing our QNet actually captures some general capability of long term value modeling.  On the other hand, if we modify the model weights of the language policy, e.g., using RL to optimize the LLM policy with the q function, this will make the language agent overfit more to this task distribution rather than generalizable to other environments.
>
> > Question on MDP design using the full observations and actions as current states
>
> We would like to argue that most of the language agent works such as Agent Q[1], language agents are using the full states and actions to serve as the current state. This approach arises from the fundamental differences between language agent tasks and traditional reinforcement learning (RL) environments. In language agent tasks, the observations received from the environment at each step often lack sufficient context to fully represent the state of the environment. This complete interaction history is critical for enabling the agent to reason effectively and make informed decisions.
>
> >  Experimental Results
>
> Thanks for the comments. We would like to argue that the current improvement on the webshop is fairly good enough. For example, in ETO[2], Agent Lumos[3], and Agent Q[1], they also get around 7~10% performance gain on the WebShop, which already is a good enhancement compared with baselines on the webshop in the language agent domain. Also, we want to claim that tree search algorithm, given the limited resources in the academia, running one iteration from fine-tuning, self-generation, training QNet and using QNet to guide inference already takes three to four days if using 4*A6000. Considering that ALFWorld，ScienceWorld takes long steps to finish, it might take more than one week even if we have one node of A100 to finish one iteration of experiments mentioned above. We are conducting experiments on new agent tasks and making every effort to update the results before the rebuttal deadline. We sincerely thank the reviewer for the understanding.
>
>
> References
>
> [1] Putta et al. (2024) Agent Q: Advanced Reasoning and Learning for Autonomous AI Agents
>
> [2] Song et al. (2024) Trial and Error: Exploration-Based Trajectory Optimization for LLM Agents
>
> [3] Yin et al. (2023) Agent Lumos: Unified and Modular Training for Open-Source Language Agents

---

> > ### Comment · Reviewer_gDbK · 2024-11-25
> > **Thanks for replying**
> >
> > Thank you for the revisions. I still have some concerns regarding the *depth of exploration*. Why does a depth of 7 result in lower trajectory quality compared to settings with depths of 3 or 5? Additionally, given that the empirical evaluation heavily relies on the *WebShop* environment, I encourage the authors to include a mention of *WebShop* or *web navigation* in the introduction section. After reviewing your future clarifications and revised version, I will be inclined towards a *weak accept*, as I would like to see more diversity in the evaluation scenarios, even if they are relatively simple.

---

> > > ### Author Response · Authors · 2024-11-25
> > >
> > > Thank you for your valuable feedback! We would like to address your concerns as follows:
> > >
> > > > The lower trajectory quality observed with exploration depth of 7
> > >
> > > The WebShop SFT dataset contains expert trajectories with an average length of approximately 5. When the exploration depth exceeds this average, such as reaching 7, it is more likely to expand the nodes on low-quality branches with depth levels of at least 7. This additional exploration of low quality nodes may further introduce trajectories of low quality.
> > >
> > > > Introduction revision
> > >
> > > Thanks for the suggestion on further clarifying our experimental setup in introduction. Following your advice, we have revised the introduction to explicitly state that we mainly conduct evaluation in web navigation tasks.
> > >
> > > Thank you again for your valuable feedback. We are happy to address any further questions you might have.

---

> > > > ### Comment · Reviewer_gDbK · 2024-11-25
> > > >
> > > > Thanks for your revisions, I increased my score.

---

> > > > > ### Author Response · Authors · 2024-11-25
> > > > >
> > > > > Thank you so much for your positive feedback! We truly appreciate the time you took to provide detailed comments and discuss with us!

---

### Official Review · Reviewer_eNPo · 2024-11-03

**Soundness:** 2
**Presentation:** 1
**Contribution:** 2
**Rating:** 3
**Confidence:** 5

**Summary:**

This paper study a critical problem in agent scenarios: the absence of process rewards for each intermediate step. The authors propose an approach that involves three main steps: 1. collecting a large number of interaction trajectories by constructing a reasoning tree; 2. using Bellman's equation to estimate the expected Q of each step; and 3. training a QNet to estimate the process reward Q values for states and actions. The experiments validate the effectiveness of QNet in providing intermediate step rewards during both training and reasoning, demonstrating its ability to offer process-based guidance and improve agent performance compared to trajectory-based rewards.

**Strengths:**

This paper is well-motivated. The lack of process rewards is a significant challenge in agent tasks. The proposed method reduces the costs associated with obtaining high-quality data annotations.

Considering training step-level verifiers is demonstrated to be effective in LLM reasoning tasks, applying the approach to agent tasks is of great significance.

**Weaknesses:**

The paper is poorly written, lacking many essential explanations and details. For example, in Section 4.4, the authors state, "we also introduce augmenting action diversity with perturbation during this stage, which is realized by prompting LLM to paraphrase the task description," yet they do not provide any discussion on prompt implementation or examples of action diversity. The termination condition for the tree construction process is not clarified, and several definitions, such as $C_t$ in Equation (4), are missing.
The experimental setup also deeds further clarification. Some critical hyper-parameters, such as the discount factor $\gamma$ for extracting Q-values is not listed. The metrics for performance in Table 1 are not explained. The sampling number per step of Q* Agent-I is not provided.
For visualization, the horizontal axis in Fig. 3(a) should indicate the number of sampled actions. Additionally, Fig. 3(b) lacks a label for the ordinate.
Overall, there are too many issues with the paper to list them all here.

The versatility of the method needs to be verified in more agent tasks.

Equation (3) is not the cross-entropy loss.

Line 415: Table 2 is not organized into three sections.

**Questions:**

In Section 5.5, how were the "Averaged reward" and "Reward" obtained during inference? I believe the average step-level reward is unavailable during inference.

What the advantage of utilizing bellman equation to estimate Q-values compared to widely-used MCTS.

The authors state that the average depths of tree searching for agent tasks are large. Any specific statistics? The authors in [1] state that the average number of steps for Webshop is 6.8.

Reference:
[1] Putta P, Mills E, Garg N, et al. Agent q: Advanced reasoning and learning for autonomous ai agents[J]. arXiv preprint arXiv:2408.07199, 2024.

---

> ### Author Response · Authors · 2024-11-22
> **Author Response to Reviewer eNPo**
>
> We express our great appreciation for the reviewer's time on reading our paper and giving us insightful suggestions. We address your concerns below.
>
> > W1、3、4 Paper writing needs to be refined.
>
> Thanks for the detailed reviews and valuable suggestions to refine our paper! We have revised our paper according to your advice.
>
> > W2 The versatility of the method needs to be verified in more agent tasks.
>
>  Thanks for the advice. Currently, we are conducting experiments on new agent tasks requiring larger exploration space. Due to the increased trajectory length in these tasks and the need to adjust hyperparameters to adapt to the new tasks, it takes approximately one week to complete a single run on eight A00 GPUs. We are doing our best to obtain these results as an academic lab with limited computational resources, and we will make every effort to update them before the rebuttal deadline. Meanwhile, we address the other points first. We sincerely thank the review for the understanding.
>
> > Q1 In Section 5.5, how were the "Averaged reward" and "Reward" obtained during inference? I believe the average step-level reward is unavailable during inference.
>
> We apologize for any confusion caused by our explanation. We would like to clarify that the process rewards are not obtained during inference but are instead computed after Stage 2 in our framework. Specifically, we first construct a reasoning tree during the self-exploration phase in Stage 2. The recorded rewards for each node in the tree are then utilized to construct various process rewards and train process reward models. The results shown in the figure correspond to the Q*Agent-ST (self-training) setup. We have refined our writing in this section and added more explanations.
>
> > Q2 What is the advantage of utilizing Bellman equation to estimate Q-values compared to widely-used MCTS?
>
> Compared with the widely-used MCTS, our method shows difference in the following perspectives:
>
> 1. The reward in the MCTS does not take account the depth of the trajectory, while the Q value function adopts a discounted value for the future outcome, which is better at modeling long-term value.
> 2. MCTS is a model-based setup, which requires the environment to output the states so that they can leverage to do backtracking in their framework during the sampling process, e.g., in AgentQ and MCTS. Our method, which adopts a Q-learning framework, is a model-free setup which does not necessarily need to know the function of how the environment outputs the future states. The sampling process is purely guided by feeding historical states and current action to the QNet.
>
> > Q3  Statistics of agent tasks
>
> We adopt the open-sourced expert trajectory datasets in ETO[1]. The average turns of interactions of expert trajectories in WebShop, ScienceWorld, and ALFWorld are 4.9, 14.4, and 10.1, respectively. To construct an effective reasoning tree, the number of trajectories that need to be explored grows exponentially with the depth of the tree. This further emphasizes the importance of designing effective node expansion strategy with proper tree pruning.
>
> References
>
> [1] Song et al. (2024) Trial and Error: Exploration-Based Trajectory Optimization for LLM Agents

---

> > ### Comment · Reviewer_eNPo · 2024-11-26
> > **Comparison with MCTS**
> >
> > > 1. The reward in the MCTS does not take account the depth of the trajectory, while the Q value function adopts a discounted value for the future outcome, which is better at modeling long-term value. 2. MCTS is a model-based setup, which requires the environment to output the states so that they can leverage to do backtracking in their framework during the sampling process, e.g., in AgentQ and MCTS. Our method, which adopts a Q-learning framework, is a model-free setup which does not necessarily need to know the function of how the environment outputs the future states. The sampling process is purely guided by feeding historical states and current action to the QNet.
> >
> > 1. Are there ablation experiments to support the importance of the discount factor?
> >
> > 2. That is not the point. As far as I am concerned, your proposed method also requires the environment to output the states for backtracking during constructing the reasoning tree. Besides, MCTS can also be used to collect Q values for training the model-free QNet. The author should directly compare with MCTS to demonstrate the proposed Q-value estimation strategy.

---

> > > ### Author Response · Authors · 2024-12-03
> > >
> > > > Are there ablation experiments to support the importance of the discount factor?
> > >
> > > Yes, we have conducted ablation experiments to evaluate the impact of the discount factor on performance. In our paper, we compared different methods of computing the process reward under the self-training setup, as shown in Figure 3. The results are as follows: Average Reward (65.4), Q-value (66.4), and Reward (64.7).
> > >
> > > We have also conducted an additional ablation study on the discount factor. The results under the same search budget are in the following table. The discount factor used in our paper is 0.9.
> > >
> > > |  Method| Discount Factor $\gamma$  | WebShop Reward|
> > > |---------|---------|---------|
> > > |Llama-2-7B-Chat-Q*Agent-I–aug |   1.0   |  70.1     |
> > > |Llama-2-7B-Chat-Q*Agent-I-aug  |   0.9   |   70.3    |
> > > We can observe that $\gamma=0.9$ provides a slight advantage over $\gamma=1.0$. This minor improvement is due to the limited difference in trajectory length of WebShop. In WebShop, the maximum number of self-exploration steps is set to 5. Given the nature of the task, the trajectory length is typically at least 3 steps (comprising an initial keyword search, intermediate item review steps, and a final purchase action), with the longest trajectory being 5 steps. As a result, the discount factor's influence on performance is relatively modest, since the range of steps over which discounting is applied is limited.
> > >
> > > > Comparison with MCTS
> > >
> > > We would like to compare with the results of a concurrent work[1] to show the effectiveness of our methods. [1] use data collected from MCTS to train a Q-value network to guide the action sampling process of language agents. We both experimented on WebShop and used the same open-sourced training set from ETO and the same evaluation metric of average trajectory rewards on the test set. In their experiments, even though [1] use a much stronger base model (GPT-4-turbo), the best evaluation result on WebShop is only 0.64, while our Q*Agent achieves the best result of 0.726 with Llama-2-7B-Chat. We believe this comparison can show the superiority of our proposed tree-search process compared with MCTS.
> > >
> > > [1] Zhai et al. (2024) Enhancing Decision-Making for LLM Agents via Step-Level Q-Value Models

---

> ### Comment · Reviewer_eNPo · 2024-11-26
>
> I appreciate the authors' efforts in providing responses; however, not all of my concerns have been adequately addressed. The paper still lacks clarity in several key areas. For instance, the termination condition for building the inference tree and the temperature used for sampling actions are not provided.
> Additionally, the ablation studies are insufficient to convincingly demonstrate the effectiveness of each proposed component.
>
> While the paper is well-motivated and tackles a significant challenge in the domain of LLM agents, I am maintaining my current rating due to the unsatisfactory writing quality, the absence of these critical details, the lack of robust evidence supporting the effectiveness of each component, and the unavailability of the source code. These factors collectively impact the paper's quality and overall completeness.

---

> ### Author Response · Authors · 2024-12-02
>
> Thank you for your precious feedback. We answer your concern and comments as follows:
>
> **(1) the termination condition for building the inference tree and the temperature used for sampling actions are not provided**
>
> We have added these details in the revised paper (Line 390~393).  “During sampling process, the environment will
> give termination signal after certain action “Click” or achieve the maximum steps set in advance.
> Specifically, we set the maximum as 5 for WebShop during self-generation, Q-guided generation and
> evaluation on the test set.” and we set temperature as 0.7 (Table3 in page16).
>
> **(2) comparison with MCTS**
>
> We would like to adopt the results from Zhao et al. (2024)[1], which is our concurrent work, to show the effectiveness of our methods. Note that they are using data collected from MCTS to train a Q-value network to guide the action sampling process of language agents. In their experiments, even though they are using a much stronger base-model (GPT-4-turbo), the best  average outcome reward for WebShop is only 0.64, which is significantly lower than our best results (0.726) with Llama-2-7B. We believe this observation can prove the superiority of our proposed tree-search process compared with MCTS.
>
> **(3) robust evidence supporting the effectiveness of each component**
>
> As our key design, we actually have quantitative and qualitative analysis of our proposed QNet compared with other reward function designs. To better showcase the robustness of our proposed method, we conducted experiments on different policies, e.g., models with different levels of behavior cloning, and models with different numbers of parameters.
> We have done ablations using different process rewards and its inference efficiency (Figure 3.), and we add experiments using a new SFT dataset filtering out trajectories with rewards higher than 0.5 from the original dataset. The results are below. From the table, we can observe that our $Q^{\star}$Agent-I-aug can greatly enhance agent inference by providing effective process reward guidance with perturbation augmented action. Besides,  $Q^{\star}$Agent-ST has better performance than ETO, demonstrating its effectiveness in generating high-quality data for self-training. Overall, $Q^{\star}$Agent still performs exceptionally well even when the SFT dataset is with poor quality.
> | **Method**       | **Reward** |
> |-----------------|-----------|
> | SFT             | 21.7      |
> | Best-of-N       | 22.1      |
> | Best-of-N-aug       | 26.8     |
> | Q*Agent-I       | 42.6      |
> | Q*Agent-I-aug   | 45.9      |
> | Q*Agent-ST      | **47.9**      |
> | ETO             | 47.0      |
>
> To give a stronger support to the empirical results, we run the experiments on the sciworld in a limited time. We have achieved better performance than SFT, PPO, BoN, RFT, and comparable to ETO.
>
>
> | Method                      | ScienceWorld |
> |-----------------------------|--------------|
> | GPT-4                       | 64.4         |
> | GPT-3.5                     | 13.0         |
> | LLama-2-7B-Chat             | 3.1          |
> | LLama-2-7B-Chat-SFT         | 53.0         |
> | LLama-2-7B-Chat-BoN         | 57.6         |
> | LLama-2-7B-Chat-RFT         | 54.3         |
> | LLama-2-7B-Chat-PPO         | 51.7         |
> | LLama-2-7B-Chat-ETO         | 65.0         |
> | Q*Agent-I                   | 63.9         |
>
>
> As shown in the table, our q-guided exploration has already achieved good enough performance on both Webshop and Sciworld. We believe compared with PPO and DPO-based methods like ETO, our method is simpler and more flexible without further modifying the original model weights. In practice, we found tuning DPO and PPO is hard (the parameter is sensitive) and cost a lot of compute.
>
> Since ETO does not release their DPO checkpoints, we apply our checkpoints on other well-trained checkpoints like llama-13B-instruct to see whether it can improve the checkpoints.
> | Method               | ScienceWorld |
> |----------------------|--------------|
> | LLama-2-13B-SFT     | 51.4         |
> | Q*Agent-I (13B)     | 61.3         |
>
>
> **(4) the unavailability of the source code**
>
> We will definitely release the code upon acceptance. We want to claim that it is not required to provide the source code at the submission stage.
>
> Thank you so much for your feedback! We truly appreciate the time you took to provide detailed comments and discuss with us!
>
> References:
> [1] Zhai et al (2024). Enhancing Decision-Making for LLM Agents via Step-Level Q-Value Models.

---

### Official Review · Reviewer_7xr3 · 2024-11-06

**Soundness:** 2
**Presentation:** 3
**Contribution:** 1
**Rating:** 5
**Confidence:** 4

**Summary:**

The paper introduces Q*Agent, a novel approach of building language agents through learning a Q function and use it to select action. Experimental results on the WebShop benchmark show that QAgent achieves strong performance and efficiency gains compared to baseline methods.

**Strengths:**

1. This paper introduced learning a Q function, which provides step-wise feedback instead of outcome-based rewards for the language agents.
2. It proposed two ways of leveraging Q functions, one for selecting actions in inference and one for filtering data in training/finetuning.

**Weaknesses:**

1. The experimental study is limited on one domain. It is unclear how effective the proposed method on other types of agent task, or even on other types of websites.

2. The experimental study lacks discussion about related work. For inference-time self-improvement, the experiment only compared the proposed method with best-of-N and ignores a large body of related work on language agent. I think the following work need to be discussed and compared with the proposed method.
 - Some fundamental work on self-improvement on language agent, for example the Reflexion and LATS methods.
 - Some work use a trained model (not per-step) to provide feedback for self-improvement at the inference time. E.g. "Autonomous Evaluation and Refinement of Digital Agents Jiayi Pan, Yichi Zhang, Nicholas Tomlin, Yifei Zhou, Sergey Levine, Alane Suhr"
 - Some work use per-step feedback in self-improvement, without finetuning a separate Q functions, in language agent tasks. E.g. "Tree Search for Language Model Agents Jing Yu Koh, Stephen McAleer, Daniel Fried, Ruslan Salakhutdinov"

When compare with methods without finetuning a new model, the paper also needs to justify the cost of finetuning additional model.

3. It seems the base approach in the experimental study include common prompting methods such as CoT, few-shot, ReAct etc. It is unclear if the benefit of the proposed method is orthogonal, or covered by these basic prompting approaches for language agent.

**Questions:**

1. Can you replace the illustrative example in Figure 2 with an example from (or more close to) the experimental study in this paper. This prevent it from over-claiming or misleading the applicability of the proposed method.
2. Please use $Q^\star$ instead of $Q^*$ to denote the optimal state-action value function.

---

> ### Author Response · Authors · 2024-11-22
> **Author Response to Reviewer 7xr3**
>
> We would like to express our thanks to the reviewer for the detailed reviews! We answer the questions below.
>
> > W1 Experiments on other agent tasks
>
>  Thanks for the advice. Currently, we are conducting experiments on new agent tasks requiring larger exploration space. Due to the increased trajectory length in these tasks and the need to adjust hyperparameters to adapt to the new tasks, it takes approximately one week to complete a single run on eight A100 GPUs. We are doing our best to obtain these results as an academic lab with limited computational resources, and we will make every effort to update them before the rebuttal deadline. Meanwhile, we address the other points first. We sincerely thank the review for the understanding.
>
> >  W2 Adding suggested baselines
>
> We appreciate your insightful feedback regarding the inclusion of related works on inference-time self-improvement in language agents. We also acknowledge the importance of positioning our work relative to existing works. However, we respectfully argue that the baselines you have mentioned (e.g. LATS, Reflexion) rely on an external LLM or VLM to provide verbal feedback rather than scalar rewards. This places them in a fundamentally different category of work. Besides, the focus of our work is fundamentally different as we aim to achieve self-improvement with minimal reliance on external resources.
>
> > W3 Further explanation on the prompting setting
>
> We would like to clarify that the benefits of ReAct in the SFT dataset and 1-shot evaluation setup are orthogonal to our method. For fair comparison, all the experimental results in the table are evaluated under the same 1-shot prompt, and all the training-based methods including ETO, RFT, PPO, QAgent-ST are based on the same SFT dataset in ReAct style.
>
> > Q1 Replace the illustrative example in Figure 2 with an example more close to the experimental study in this paper.
>
> Thanks for the suggestion. We have changed the example to one more close to the WebShop environment.
>
> >  Q2 Use $Q^\star$ instead of Q* to denote the optimal state-action value function.
>
> Thanks for the advice. We have revised the notion in our paper to denote the estimated optimal state-action value function.

---

> > ### Author Response · Authors · 2024-12-02
> > **Experiments on more agent tasks and different policy base models**
> >
> > We want to claim that our q-guided sampling method can be actually applied on top of any checkpoints, including the models using training-based alignment algorithm like PPO and DPO like ETO as a plug-and-play module. In other words, we can use our trained QNet to guide the exploration for other well-trained checkpoints.
> > To give a stronger support to the empirical results, we run the experiments on the sciworld in a limited time. We have achieved better performance than SFT, PPO, BoN, RFT, and comparable to ETO.
> >
> > | Method                      | ScienceWorld |
> > |-----------------------------|--------------|
> > | GPT-4                       | 64.4         |
> > | GPT-3.5                     | 13.0         |
> > | LLama-2-7B-Chat             | 3.1          |
> > | LLama-2-7B-Chat-SFT         | 53.0         |
> > | LLama-2-7B-Chat-BoN         | 57.6         |
> > | LLama-2-7B-Chat-RFT         | 54.3         |
> > | LLama-2-7B-Chat-PPO         | 51.7         |
> > | LLama-2-7B-Chat-ETO         | 65.0         |
> > | Q*Agent-I                   | 63.9         |
> >
> >
> > As shown in the table, our q-guided exploration has already achieved good enough performance on both Webshop and Sciworld. We believe compared with PPO and DPO-based methods like ETO, our method is simpler and more flexible without further modifying the original model weights. In practice, we found tuning DPO and PPO is hard (the parameter is sensitive) and cost a lot of compute.
> >
> > Since ETO does not release their DPO checkpoints, we apply our checkpoints on other well-trained checkpoints like llama-13B-instruct to see whether it can improve the checkpoints.
> > | Method               | ScienceWorld |
> > |----------------------|--------------|
> > | LLama-2-13B-SFT     | 51.4         |
> > | Q*Agent-I (13B)     | 61.3         |
> >
> > Theoretically, our method is a simple yet novel way to model process rewards using q-learning approach while ETO uses a trajectory-level reward function to do DPO. In complex agent environment, our method is expected to better capture long-term value for each step, while trajectory-level reward function is not explicitly designed for that.

---

> > > ### Author Response · Authors · 2024-12-02
> > > **More baselines**
> > >
> > > Also, we have added Reflexion and LATS as the baseline in Table1 (please refer to Line 333 - 334), where their methods rely on ChatGPT with ReAct prompting.
> > > | Method                              | WebShop |
> > > |-------------------------------------|---------|
> > > | GPT-4                               | 63.2    |
> > > | GPT-3.5-Turbo                       | 62.4    |
> > > | Reflexion (Shinn et al., 2023)      | 64.2    |
> > > | LATS (Zhou et al., 2024)            | 75.9    |
> > > | Llama-2-7B-Chat                     | 17.9    |
> > > | Llama-2-7B-Chat + SFT               | 63.1    |
> > > | Llama-2-7B-Chat + RFT               | 63.6    |
> > > | Llama-2-7B-Chat + Q⋆Agent-ST        | 66.4    |
> > > | Llama-2-7B-Chat + PPO               | 64.2    |
> > > | Llama-2-7B-Chat + ETO               | 67.4    |
> > > | Llama-2-7B-Chat + Best-of-N         | 65.3    |
> > > | Llama-2-7B-Chat + Best-of-N-aug     | 68.4    |
> > > | Llama-2-7B-Chat + Q⋆Agent-I         | 65.5    |
> > > | Llama-2-7B-Chat + Q⋆Agent-I-aug     | 72.6    |

---

### Official Review · Reviewer_kPgp · 2024-11-07

**Soundness:** 2
**Presentation:** 2
**Contribution:** 2
**Rating:** 5
**Confidence:** 2

**Summary:**

This paper proposes a method to train a language agent, enabling it to handle complex tasks. To this end, the authors initialize the language agent via performing behavioral cloning on the collection of expert trajectories. Then, the authors utilize the supervised fine-tuned agent to explore the environment and collect trajectories. Using the collected trajectories, QNet is trained via Q-learning. Finally, Q-guided exploration is used to inference, and the Q\*Agent is trained using SFT dataset and Q-guided trajectories. Based on this training steps, Q\*Agent achieves state-of-the-art performance.

**Strengths:**

The authors propose a method to train a language agent that achieves state-of-the-art performance.

**Weaknesses:**

I have some questions about this work, which are mentioned in the Questions section below.

**Questions:**

I am unsure whether I have fully understood this paper, so please let me know if I have any misunderstandings. I would be very happy to gain a complete understanding of your work.
1. As mentioned in Section 2.2, Q\*Agent appears to be closely related to Q\*(Wang et al., 2024a) and the Q-value model enhanced (Zhai et al., 2024). In my understanding, based on your summary, I am uncertain whether adding a behavioral cloning stage is an important contribution. Could you provide a more detailed explanation of your contributions compared to these previous works?
2. During the step 2 in Figure 1, Q\*Agent stops exploring a branch’s nodes if the branch yields a zero reward. However, with this strategy, I wonder if Q\*Agent disregards partially correct trajectories where only the final parts are incorrect.
3. In Q-guided self-training, why does the Q\*Agent algorithm not use the dataset collected during step 2 in Figure 1?
4. In my understanding, the “exploration” steps, steps 2 and 4 in Figure 1, seem to be closer to exploitation rather than exploration. In RL literature, exploration typically aims to gather information about unknown parts of the environment. However, in this paper, both of these “exploration” steps use the best action based on current knowledge (reward of 1 or action with max Q), which generally aligns more closely with exploitation in RL. This terminology may be confusing for researchers who familiar with RL concepts.
5. In the experimental section, there are no results for Q\*(Wang et al., 2024a) or the Q-value model enhanced (Zhai et al., 2024). Is there any reason these algorithms were omitted?
6. In the experimental section, performing SFT appears sufficient to achieve satisfactory results. Therefore, I am curious about the performance of Q\*Agent when the SFT dataset shows poor performance, in order to observe the impact of RL.
7. I am confused by the definitions of Q\*Agent-ST and Q\*Agnet-I. As I understand it, during step 4 in Figure 1, Q\*Agent-ST involves using QNet to compute Q-values for m actions and selecting the best action, while Q\*Agent-I refers to using action sampled from the agent without using QNet. If this is correct, then what is the purpose of QNet?

Minor comment:
The notations in Appendix A.3 are somewhat confusing. The notations $\mathcal{A}$, $\mathcal{Q}$, and $q_m$ should be defined formally. Additionally, in line 7, $\arg\max$ is not used properly according to its definition.

**Details Of Ethics Concerns:**

-

---

> ### Author Response · Authors · 2024-11-22
> **Author Response to Reviewer kPgp (1/N)**
>
> We appreciate the reviewer’s insightful feedback. We address your concerns below one by one.
>
> > Q1 More detailed comparison with Q*[1] and the enhanced Q-value model[2].
>
> We would like to clarify first that SFT for behavioral cloning is not our unique contribution. We use SFT to provide our language agent with capabilities to perform reasoning and actions in this environment. Also, we respectfully argue that both Q*[1] and the enhanced Q-value model[2] are concurrent works. The comparisons are listed below:
>
> 1. Q*[1] has not been tested on interactive agent tasks which are more complicated and have broader real-world applications. Also, it does not include a self-training stage to further enhance the language agents.
>
> 2. We introduce several strategies to reduce the time complexity of tree search to address the heavy searching cost in interactive agent tasks. Also, we propose using LLM to perturb the context to further boost the diversity of the exploration content.
>
> 3. For training the process reward model, unlike  the Q-value model[2] that leverages DPO, our framework adopts MSE loss to learn the Q-value. This approach is not only much easier to optimize but also more robust to the hyper-parameters.
> 4. Both Q*[1] and the enhanced Q-value model[2] have not released their code, so we cannot reproduce their methods in our setup. However, since Zhai et al.[2] also experimented on WebShop and used the same SFT dataset in ETO[3] as $Q^{\star}$Agent, we gave a concise comparison in L424-427 of our paper: Zhai et al.[2] used Llama-3.1-8b-instruct as their base agent and achieved a final reward of **60**. Our $Q^{\star}$Agent used llama2-7b-chat as base agent and  $Q^{\star}$Agent-I-aug achieved a higher final reward of **72.6**.
>
> >  Q2 Whether pruning zero-branches will disregard partially correct trajectories
>
> First we would like to clarify that our approach does not prune or discard branches with a final reward of zero that have already been explored, which could be partially correct trajectories. Designing the pruning strategy is intended to optimize exploration focusing on expanding branches that are more likely to yield non-zero rewards within a limited time. Therefore, while we cease further expansion at nodes deep within a trajectory that culminates in zero reward, we retain the trajectory up to that point for analysis and potential use in subsequent training iterations.
>
> To provide further clarity on how $Q^{\star}$Agent constructs the reasoning tree, we have added Algorithm 3 in Appendix A.4. of our revised version for your reference.
>
> >  Q3 In Q-guided self-training, why does the $Q^{\star}$Agent algorithm not use the dataset collected during step 2 in Figure 1?
>
> We do not use the trajectories in step 2 because there are many step-level overlaps between the self-explored trajectories used to train the QNet and Q-guided explored trajectories. Also, we believe self-training requires data with higher quality, and q-guided explored trajectories have higher quality than the self-explored one, so we do not use the data collected during step 2 for the current version.
>
> >  Q4 The 'exploration' terminology may be confusing
>
> Thanks for pointing this out. We revised the terminology used in step2 and step4 to “self-guided generation” and “Q-guided generation” respectively.

---

> ### Author Response · Authors · 2024-11-22
> **Author Response to Reviewer kPgp (2/N)**
>
> >  Q5 Is there any reason results of Q*[1] and the enhanced Q-value model[2] were omitted?
>
> As we have stated in answer to Q1, due to the absence of an open-source codebase for the two mentioned concurrent works, we are unable to reproduce their results within our setup and data framework. We provide a concise comparison with Zhai et al. (2024), as we both involve experiments on WebShop using the same SFT dataset from the ETO[3]. This comparison is detailed in Lines 424-427 of our paper: Zhai et al.[2] used Llama-3.1-8b-instruct as their base agent and achieved a final reward of **60**. Our $Q^{\star}$Agent used llama2-7b-chat as base agent and  $Q^{\star}$Agent-I-aug achieved a higher final reward of **72.6**.
>
> >  Q6 The performance of $Q^{\star}$Agent when the SFT dataset shows poor performance
>
> Thanks for the insightful suggestion! We add experiments using a new SFT dataset filtering out trajectories with rewards higher than 0.5 from the original dataset. The results are below. From the table, we can observe that our $Q^{\star}$Agent-I-aug can greatly enhance agent inference by providing effective process reward guidance with perturbation augmented action. Besides,  $Q^{\star}$Agent-ST has better performance than ETO, demonstrating its effectiveness in generating high-quality data for self-training. Overall, $Q^{\star}$Agent still performs exceptionally well even when the SFT dataset is with poor quality.
>
> | **Method**       | **Reward** |
> |-----------------|-----------|
> | SFT             | 21.7      |
> | Best-of-N       | 22.1      |
> | Best-of-N-aug       | 26.8     |
> | Q*Agent-I       | 42.6      |
> | Q*Agent-I-aug   | 45.9      |
> | Q*Agent-ST      | **47.9**      |
> | ETO             | 47.0      |
>
> >  Q7 Further explanation on the setting of  $Q^{\star}$Agent-ST and  $Q^{\star}$Agent-I
>
> Thanks for the question. Both $Q^{\star}$Agent-ST and $Q^{\star}$Agent-I use QNet to choose actions with the highest Q-value at each generation step. $Q^{\star}$Agent-I is using QNet during the searching process at each step, using Q(current_state, current_action) to predict the q-value for each action and to select the action with the highest q-value after sample three actions for each step. $Q^{\star}$Agent-ST also uses QNet to do such step-wise search and collect sampled trajectories to train the language agent.
>
> >  Minor revisions on notations in the Algorithm 2 of Appendix A.3.
>
> Thanks for pointing this out. We have rewritten notations and restructured the Algorithm 2 in Appendix A.3 to provide a more clear explanation. We kindly invite the reviewer to refer to our revised paper.
>
>
>
> References
>
> [1] Wang et al. (2024) Q*: Improving Multi-step Reasoning for LLMs with Deliberative Planning
>
> [2] Zhai et al. (2024) Enhancing Decision-Making for LLM Agents via Step-Level Q-Value Models
>
> [3] Song et al. (2024) Trial and Error: Exploration-Based Trajectory Optimization for LLM Agents

---

> ### Comment · Reviewer_kPgp · 2024-11-26
>
> Thank you for your detailed response.
>
> However, I am still confused about the definitions of -ST and -I.
>
> Additionally, it is not clear that Q\*Agent performs better than other baselines, both empirically and theoretically, including ETO.
>
> As a result, I am on the borderline and would not be surprised if this work gets accepted, but I will maintain my current score.

---

> > ### Author Response · Authors · 2024-12-02
> > **Additional experiments and explanations**
> >
> > Thank you for your comment.
> >
> > (1) **The definitions of -ST and -I**
> >
> > To give a more clear explanation, -ST is the method that needs to modify the weights of the policy model by finetuning on the original policy using the explored dataset (q-guided exploration). -I is the method that only needs to train the QNet to the guide the policy model without need to do further exploration and retrain on the original policy. We have revised the corresponding part in the paper and will give more clear explanations in the final version.
> >
> > (2) **The advantages of Q*Agent against other baselines**
> >
> > First of all, we want to claim that our q-guided sampling method can be actually applied on top of any checkpoints, including the models using training-based alignment algorithm like PPO and DPO like ETO as a plug-and-play module. In other words, we can use our trained QNet to guide the exploration for other well-trained checkpoints.
> > To give a stronger support to the empirical results, we run the experiments on the sciworld in a limited time. We have achieved better performance than SFT, PPO, BoN, RFT, and comparable to ETO.
> >
> > | Method                      | ScienceWorld |
> > |-----------------------------|--------------|
> > | GPT-4                       | 64.4         |
> > | GPT-3.5                     | 13.0         |
> > | LLama-2-7B-Chat             | 3.1          |
> > | LLama-2-7B-Chat-SFT         | 53.0         |
> > | LLama-2-7B-Chat-BoN         | 57.6         |
> > | LLama-2-7B-Chat-RFT         | 54.3         |
> > | LLama-2-7B-Chat-PPO         | 51.7         |
> > | LLama-2-7B-Chat-ETO         | 65.0         |
> > | Q*Agent-I                   | 63.9         |
> >
> >
> > As shown in the table, our q-guided exploration has already achieved good enough performance on both Webshop and Sciworld. We believe compared with PPO and DPO-based methods like ETO, our method is simpler and more flexible without further modifying the original model weights. In practice, we found tuning DPO and PPO is hard (the parameter is sensitive) and cost a lot of compute.
> >
> > Since ETO does not release their DPO checkpoints, we apply our checkpoints on other well-trained checkpoints like llama-13B-instruct to see whether it can improve the checkpoints.
> > | Method               | ScienceWorld |
> > |----------------------|--------------|
> > | LLama-2-13B-SFT     | 51.4         |
> > | Q*Agent-I (13B)     | 61.3         |
> >
> > Theoretically, our method is a simple yet novel way to model process rewards using q-learning approach while ETO uses a trajectory-level reward function to do DPO. In complex agent environment, our method is expected to better capture long-term value for each step, while trajectory-level reward function is not explicitly designed for that.

---

### Author Response · Authors · 2024-11-22

We sincerely thank all the reviewers for their time and valuable feedback. We have carefully addressed all the comments and refined our paper accordingly, submitting a revised version for your consideration. If there are any further questions or concerns, we would be delighted to continue the discussion. Thank you once again for your thoughtful reviews and support!

---

### Author Response · Authors · 2024-12-04

We sincerely appreciate the valuable feedback from all reviewers. We have addressed each concern thoroughly in a point-by-point manner, incorporating new experimental results and revising the manuscript to reflect updates to the methodology, experiments, and appendix.
Please feel free to reach out with any further questions or concerns regarding the paper. We would be glad to provide additional clarifications and make further improvements as needed.

Our revised manuscript includes the following major new analyses and experiments:
 - Adding more experiments to validate the effectiveness of the proposed method. (Reviewer kPgp, 7xr3)
 - Adding self-improvement baselines including Reflexion and LATS (Reviewer 7xr3)
 - Difference with MCTS and other related work (Reviewer eNPo, gDbK, kPgp)
 - Writing improvement. (Reviewer eNPo)

---

### Meta-Review · Area_Chair_5pFc · 2024-12-21

**Metareview:**

This paper develops a procedure that collects trajectories from a chatbot-like agent to construct a reasoning tree, builds a value function to estimate the utility of each action in this tree, and uses this procedure for Q-function guided generation/action selection on web-navigation tasks. The discourse with the reviewers revolved around clarifying questions about the approach, relationship to recent/concurrent on these problems that is rather similar, and the effectiveness of learning a value function for web navigation tasks where the action space is very large.

Although some of these concerns are alleviated after the author responses, the key ones are not, e.g., the incremental nature of these ideas in light of existing work on this topic (Reflexion, ETO etc.), limited improvements on benchmark problems. It would also be useful to demonstrate more thorough experimental evidence of the utility of this approach, e.g., how does this approach quantitatively address the fact that there are a lot of actions to choose from while fitting, or using the value function? For these reasons, I do not recommend that this paper is accepted.

**Additional Comments On Reviewer Discussion:**

Reviewer kPgp: there were a number of clarification questions about the approach, which the authors have addressed carefully, e.g., by conducting new experiments on the SFT dataset. There were no comments made by the reviewer either positive or negative.

Reviewer 7xr3: suggested a number of connections to existing work on tree-search based exploration, learning the per-step reward model etc. They wanted the authors to discuss results on more domains, but this was understandably difficult to pull off during the rebuttal period. It is important to check the standard deviation of the numerical results reported in the new experiments to compare against baseline procedures.

Reviewer eNPo wanted better explanations of the approach and analysis of the hyper-parameters. After a discourse, the eventual summary of the reviewer was that the evidence in the paper is not sufficient to ascertain the effectiveness of the approach. The authors have provided explanations for these questions including supporting experiments of their own and from existing papers. I discounted low score (3/10) of Reviewer eNPo in the decision.

Reviewer gDbK had concerns about the positioning of the paper (MCTS, Web Shop style evaluation), and some more comments on the length of the trajectories. The authors have addressed these concerns satisfactorily and the reviewer updated their score.

---

### Decision · Program_Chairs · 2025-01-22

Reject